# Recent Advances in the Use of Vitamin D Organic Nanocarriers for Drug Delivery

**DOI:** 10.3390/biom14091090

**Published:** 2024-08-30

**Authors:** Ioanna Aggeletopoulou, Maria Kalafateli, Georgios Geramoutsos, Christos Triantos

**Affiliations:** 1Division of Gastroenterology, Department of Internal Medicine, University Hospital of Patras, 26504 Patras, Greece; iaggel@upatras.gr (I.A.); giorgosgeramoutsos@gmail.com (G.G.); 2Department of Gastroenterology, General Hospital of Patras, 26332 Patras, Greece; mariakalaf@hotmail.com

**Keywords:** nanocarriers, vitamin D, VDR, nanoparticles, encapsulation, organic carriers, drug delivery

## Abstract

Nanotechnology, now established as a transformative technology, has revolutionized medicine by enabling highly targeted drug delivery. The use of organic nanocarriers in drug delivery systems significantly enhances the bioavailability of vitamins and their analogs, thereby improving cellular delivery and therapeutic effects. Vitamin D, known for its crucial role in bone health, also influences various metabolic functions, such as cellular proliferation, differentiation, and immunomodulation, and is increasingly explored for its anticancer potential. Given its versatile properties and biocompatibility, vitamin D is an attractive candidate for encapsulation within drug delivery systems. This review provides a comprehensive overview of vitamin D synthesis, metabolism, and signaling, as well as its applications in customized drug delivery. Moreover, it examines the design and engineering of organic nanocarriers that incorporate vitamin D and discusses advances in this field, including the synergistic effects achieved through the combination of vitamin D with other therapeutic agents. By highlighting these innovations, this review provides valuable insights into the development of advanced drug delivery systems and their potential to enhance therapeutic outcomes.

## 1. Introduction

Vitamin D is a fat-soluble nutrient that critically contributes to diverse physiological processes within the human body. The role of vitamin D as a prohormone is well established. It serves as a precursor to its active metabolite, calcitriol, which is a lipophilic seco-steroid hormone [1]. Its function is integral to many physiological processes, extending well beyond its traditional association with the regulation of calcium and phosphorus metabolism, and, thus, with bone remodeling [1,2,3]. For example, vitamin D critically participates in the regulation of immune system responses, as calcitriol enhances the pathogen-fighting abilities of monocytes and macrophages and modulates inflammatory responses [3,4,5]. Moreover, the binding of calcitriol to its receptor, vitamin D receptor (VDR), in various tissues, results in the regulation of the expression of genes involved in cell growth (e.g., *BCL2*, *cyclin D1*), cell differentiation (e.g., *caudal type homeobox 2*, *CDX2*), immune function (e.g., *interleukin-10* or *nuclear factor kappa* B), and inflammation (e.g., *tumor necrosis factor-alpha*) [3,4,5]. This has generated significant interest in the potential pathogenic effects of vitamin D on the development and clinical course of autoimmune diseases [6,7,8]. Emerging research suggests that vitamin D might have anticancer activity through the modulation of cell growth and cell differentiation [9,10]. Recent findings also highlight the role of vitamin D on cardiovascular health, as it affects blood pressure, cholesterol levels, and endothelial function [11]. Additionally, there is growing evidence linking vitamin D to mental health, especially with regard to mood regulation and prevention of depression [12]. Vitamin D is also associated with liver disease, as it modulates immune responses, reduces inflammation, and maintains liver function, potentially mitigating disease progression [13,14,15]. Furthermore, neuroprotective effects of vitamin D have been documented, with potential benefits to neurodegenerative diseases such as Alzheimer’s and Parkinson’s disease [16]. These diverse roles underscore the therapeutic potential of vitamin D in various chronic conditions.

Vitamin D deficiency represents a significant public health issue, given its potential to lead to a range of health complications, as detailed above. On the other hand, vitamin D supplementation could potentially result in vitamin D toxicity, with hypercalcemia being its most feared adverse event [17,18]. While efforts to address deficiencies through dietary intake and supplementation are crucial, these approaches are challenging, considering issues with vitamin D’s stability, bioavailability, and targeted delivery [19,20]. Over 75% of ingested vitamin D is metabolized and excreted before its conversion into the active form or its storage in tissues, complicating the achievement of therapeutic levels. These inefficiencies not only impact the effectiveness of supplementation but also influence regulatory decisions regarding food fortification. Although fortification can mitigate deficiencies, it also increases the risk of excessive intake and potential toxicity. Addressing these challenges underscores the need for advanced delivery systems to enhance both the efficacy and safety of vitamin D therapies.

The recognition of vitamin D deficiency as a considerable health risk has led to innovations in the development of functional foods and alternative therapeutics. One promising approach involves the use of nanotechnologies to enhance the incorporation of vitamin D into food products and pharmaceutical formulations. These advancements aim to overcome challenges such as low bioavailability and ensure the preservation of vitamin D’s activity throughout processing and delivery. 

Nanocarriers in drug delivery offer significant advantages over conventional treatments by enhancing the water solubility of poorly soluble drugs and protecting them from degradation [21]. They prolong drug circulation in the bloodstream, promoting efficient accumulation at the target site [22], which can lead to reduced dosages and fewer adverse effects from both the drug and its formulation adjuvants. By utilizing passive and active targeting mechanisms, nanocarriers can accumulate precisely where needed, utilizing ligands on their surface to target specific cells [23]. This targeted approach improves drug delivery specificity, reduces systemic toxicity, and enhances therapeutic efficacy [24]. Recent studies have focused on targeting molecular markers associated with disease initiation and progression using nanoscale drug delivery systems These include various organic and inorganic nanoparticles designed for specific drug delivery goals [25,26]. Biocompatibility remains a critical challenge in the development of these systems, as it is essential to ensure they are non-toxic and do not trigger immune responses [27,28]. Other crucial factors include carrier stability and interactions with cellular membranes, which are essential for the optimization of drug delivery efficacy [29,30]. Continued refinement of these nanocarrier systems is essential for the improvement of vitamin delivery efficiency and patient compliance. 

Organic nanocarriers represent a rapidly advancing area of research with unique properties and synergies specifically relevant to vitamin D delivery. These include enhanced bioavailability, targeted delivery, and controlled release mechanisms. In contrast, inorganic platforms and natural carriers involve different material properties and design considerations. The aim of the current review is to provide a comprehensive overview of vitamin D, with a specific focus on its suitability for customized drug delivery applications. This review will delve into the design and development of organic nanocarriers that incorporate vitamin D as a pivotal component. The exploration of innovative approaches in the use of organic nanocarriers, such as liposomes, polymeric nanoparticles, and micelles, to enhance the delivery and efficacy of vitamin D, will be emphasized. Lastly, the review will discuss how these organic nanocarriers can be engineered to improve targeting, stability, and controlled release of vitamin D, thereby optimizing its therapeutic potential.

## 2. Vitamin D Uptake, Physiology and Metabolism 

The physiological effects of vitamin D are coordinated through a series of metabolic conversions that primarily take place in the skin, liver, and kidneys [2,3] (Figure 1A). 

There are two major forms of vitamin D, with different chemical structures, vitamin D2 or ergocalciferol and vitamin D3 or cholecalciferol, which contribute to their differences in function and sources [14,31]. Vitamin D2 contains a double bond between carbons 22 and 23 and a methyl group at carbon 24, whereas vitamin D3 lacks these features, which differentiate its structure [14,31]. The primary sources of vitamin D2 are plant-based, including fungi and yeast, while vitamin D3 is synthesized in the skin upon exposure to ultraviolet B (UVB) radiation from sunlight and is also found in animal-based foods such as fatty fish, liver, and egg yolks [14,31] (Figure 1A). These structural differences affect their stability and bioavailability, with vitamin D3 being more effective in increasing and maintaining overall vitamin D levels in the human body compared to vitamin D2 [14,31]. UVB (290–320 nm) sunlight activates the transformation of 7-dehydrocholesterol, a skin compound, into pre-vitamin D3 [14,32]. Pre-vitamin D3 is subjected to thermal isomerization, transforming into vitamin D3. Vitamin D does not persist in the circulation for an extended period, as it is rapidly sequestered in the adipose tissue or metabolized by the liver. During the transportation of vitamin D to the liver, it is bound to vitamin D-binding protein (DBP) (Figure 1A). Once synthesized or ingested, DBP complex experiences a two-step activation process. The first activation, which takes place in the liver, involves the conversion of DBP complex into 25-hydroxyvitamin D [25(OH)D or calcidiol] by the enzyme 25-hydroxylase (CYP2R1), which is the main circulating form of the vitamin in the serum [4,5,31] (Figure 1A). Afterwards, in the proximal tubule of the kidneys, 25(OH)D is further hydroxylated by the enzyme 1α-hydroxylase (CYP27B1), resulting in the production of the biologically active form of vitamin D, known as calcitriol or 1,25-dihydroxyvitamin D3 [1,25(OH)2D3] [4,5,31] (Figure 1A). Calcitriol enters the bloodstream, binds to DBP, and is then transported to target tissues including the intestine, kidney, and bones (Figure 1B). In these tissues, vitamin D regulates the absorption, mobilization, and reabsorption of calcium and phosphate [33]. The levels of both calcidiol and calcitriol are closely monitored by 25(OH)D 24-hydroxylase (CYP24A1), the primary enzyme responsible for vitamin D inactivation. This enzyme catalyzes hydroxylation at C-24 and C-23 positions of both calcidiol and calcitriol, leading to the formation of biologically inactive calcitroic acid, which is excreted in bile [34,35].

Although 1,25(OH)2D3 is the active form, vitamin D status is typically assessed by measuring 25(OH)D serum levels. This is preferred because 25(OH)D has a relatively long half-life and maintains stable concentrations in the bloodstream. Factors such as parathyroid hormone (PTH), calcium, phosphorus levels, and fibroblast growth factor (FGF) modulate vitamin D metabolism [36,37] (Figure 2).

PTH promotes renal expression of CYP27B1 through mechanisms that include enhancement of cAMP-dependent transcription or activation of nuclear orphan receptor NR4A2-dependent transcription. These mechanisms ultimately result in increased production of calcitriol [38] (Figure 2). Despite that elevated calcitriol levels can trigger its degradation via increased expression of *CYP24A1*, PTH maintains calcitriol levels by promoting the degradation of *CYP24A1* mRNA through stimulation of the cAMP-PKA pathway in the kidney [39]. This results in increased calcium levels, which suppress the PTH secretion in parathyroid glands, as a negative feedback loop [40]. FGF-23 can reduce serum calcitriol levels through the suppression of CYP27B1 in the kidney. Additionally, FGF-23 stimulates the expression of CYP24A1, the enzyme responsible for the degradation of both calcidiol and calcitriol, thereby further reducing its circulating levels [36,41].

## 3. Vitamin D Receptor (VDR)

The active form of vitamin D exerts its effects through binding to the VDR [42]. VDR is detectable in most human cells and vitamin D directly or indirectly modulates approximately 3–5% of the human genome; thus, the biological activity of vitamin D is extensive, encompassing actions that have the potential to mitigate the progression of numerous diseases [43]. Encoded by the *VDR* gene and belonging to the nuclear receptor superfamily of ligand-activated transcription factors, VDR functions depend on its molecular structure. For instance, the ligand-binding domain of VDR facilitates high-affinity binding with 1,25-dihydroxyvitamin D3, while the DNA-binding domain enables the receptor to bind to specific vitamin D response elements (VDREs) in the promoter regions of target genes, thus modulating transcriptional activities essential for calcium homeostasis and immune modulation [44]. Typically acting as a transcription factor, VDR impacts the expression of genes involved in diverse biological processes in response to vitamin D [45]. The binding of vitamin D to VDR can initiate various biological actions through both genomic and non-genomic pathways [46,47]. In the genomic pathway, vitamin D binds to cytosolic VDR, leading to the phosphorylation of the latter (Figure 2). Once activated, VDR forms a complex with the retinoid X receptor (RXR), which binds to specific DNA sequences known as VDREs located in the promoter regions of target genes. The VDR activation plays a critical role in either inducing or repressing the transcription of various target genes through vitamin D-mediated signaling and interactions with other transcription factors [48]. In the non-genomic pathway, calcitriol binds to membrane-bound VDR, known as 1,25D-membrane-associated rapid response steroid-binding protein (1,25D-MARRS) (Figure 2). This interplay triggers immediate alterations in cell signaling-related pathways, such as mitogen-activated protein kinase (MAPK) and calcium signaling, through direct interaction with intracellular signaling molecules that regulate specific cellular functions [49,50]. Moreover, calcitriol has VDR-independent actions through the rapid activation of signaling molecules such as phospholipase C (PLC), phosphoinositide 3-kinase (PI3K), and MAPK [51]. 

## 4. Drug Delivery Systems 

There have been numerous attempts to develop drug delivery systems (DDSs) for vitamin D or its analogues to exert both skeletal and non-skeletal effects [52]. A DDS is a method designed to introduce a therapeutic substance into the body in a controlled manner. The primary goal of a DDS is to enhance the efficacy and safety of the drug through regulation of the rate, timing, and location of its release [52]. Other DDS goals are safety, reliability, stability, cost-effectiveness, and ease of administration, without affecting drug effectiveness against different conditions [53]. The diversity in DDSs today is vast, encompassing nano- and microparticulate systems made from polymers, lipids, surfactants, and various inorganic or hybrid materials [53,54,55]. Recently, increased public interest in vitamin D has led to a surge in sales of vitamin D-related products. Various pharmaceutical forms of vitamin D supplementation, such as oily drops, capsules, and tablets, are commercially available. Given the potential therapeutic uses of vitamin D and its analogues in treating various diseases, they represent promising candidates for encapsulation into a DDS.

Nanocarriers refer specifically to DDSs that utilize nanoparticles or nanostructures to transport therapeutic agents (such as drugs or genes) to specific sites within the body [56]. These nanoparticles can vary widely in composition, size, and shape, and they are designed to improve the pharmacokinetics and pharmacodynamics of the drugs they carry [56]. Nanoparticles, due to their unique physicochemical properties, are widely used as DDSs and can be directed towards tissues via both passive and active targeting mechanisms, thus enhancing therapeutic efficacy and minimizing side effects [57] (Figure 3). 

Passive targeting utilizes the intrinsic properties of nanoparticles—such as their size, shape, and surface characteristics—that allow for them to accumulate preferably in specific tissues or organs [58]. This phenomenon, known as the enhanced permeability and retention (EPR) effect, takes advantage of the leaky vasculature and impaired lymphatic clearance found in tumors and inflamed tissues [59] (Figure 3A). Nanoparticles within a certain size range (typically 10–200 nm) can passively extravasate from blood vessels into these tissues due to their size, and once inside, they tend to accumulate due to impaired lymphatic drainage [58]. In contrast, active targeting involves the modification of nanoparticles with targeting ligands, such as antibodies, peptides, or small molecules, that can specifically bind to receptors overexpressed on the surface of target cells or tissues, often located less than 0.5 nm apart from each other [60,61]. This approach aims to enhance the specificity and selectivity of drug delivery. The active binding of nanoparticles to target cells via receptor-mediated interactions enhances cellular internalization through processes like receptor-mediated endocytosis [62] (Figure 3B). Polymers like polyethylene glycol (PEG) can coat nano-sized drug delivery vehicles, thus improving their circulation durability, reducing harmful interactions with opsonizing proteins, and limiting rapid degradation and clearance [63,64]. Active targeting strategies are designed to deliver therapeutic agents directly to the diseased cells with minimal impact on healthy tissues, reducing any systemic side effects and also improving therapeutic outcomes [65]. Techniques such as nanoparticle-based delivery systems, receptor-mediated targeting, and antibody-conjugated carriers are employed to achieve precise delivery of vitamin D to its intended site of action [66]. These methods ensure that vitamin D reaches specific tissues, such as bone or immune cells, thus optimizing its therapeutic benefits. 

While the use of nanotechnology for the treatment of various diseases is still developing, several nanocarrier-based drugs are already available, and numerous other nano-based therapeutics are currently being tested in clinical trials. Doxil and Abraxane, two of the most well-known nano-based drugs for cancer treatment, received US Food and Drug Administration (FDA) approval several years ago and have since been successfully utilized in clinical practice [67,68]. Additionally, several other cancer nanotherapeutics have been approved by the FDA and other regulatory agencies around the world [67,68]. A major challenge remains the lack of absolute selectivity. Only a small fraction (approximately 1%) of systemically administered nanoparticles reaches the diseased target site [69]. Equally concerning is the significant quantity that reaches off-target sites, leading to substantial toxicity [69]. However, most recent data comparing conventional chemotherapy and photothermal-activated nano-sized targeted drug delivery to solid tumors showed that responsive nanocarriers delivered more than 2.1 times drug to the extracellular space compared to traditional chemotherapy, thereby suppressing tumor growth for a longer duration [70].

## 5. Organic Nanocarriers

Nano-sized carriers, although originally developed for complex health issues, seem promising as delivering systems of vitamins [71,72,73]. Vitamin D, as a lipophilic agent, faces challenges regarding bioavailability due to enzymatic degradation and unsuitable chemical environments in the gastrointestinal tract [74]. Oral administration, the primary delivery method for supplements, often results in lower-than-expected intake levels due to absorption, solubility, and stability issues [75]. Factors such as underlying health conditions, diet, age, and co-medication can further affect vitamin D absorption. Taking all abovementioned into consideration, the development of novel nano-sized delivery systems to protect and enhance the systemic assimilation of vitamin D is vital. Various organic nanocarriers for encapsulating vitamin D have been developed, including lipid-based nanoparticles such as liposomes, nanostructured lipid carriers (NLCs), and lipid-core micelles, as well as polymeric nanoparticles (Figure 4). 

These carriers are designed to stabilize vitamin D, enhance its solubility, and improve its bioavailability for various therapeutic applications. The advantages and disadvantages of each delivery platform for vitamin D are summarized in Table 1.

### 5.1. Lipid-Based Nanoparticles

#### 5.1.1. Vitamin D Microencapsulation Using Liposomes

Liposomes are small, round vesicles formed by the self-organization of phospholipids, characterized by their biodegradable and biocompatible nature. These structures are classified based on their diameter, the number of bilayers, or the manufacturing method [76]. Phospholipids with hydrophobic fatty acid tails and hydrophilic heads form bilayers that can modulate fluidity and drug release [77]. Cholesterol is often included to enhance liposome stability [78]. Liposomes are considered drug carriers due to their ability to transport and release therapeutic agents in a controlled manner [79]. They protect drugs from environmental degradation, and their release can be triggered by external stimuli such as light, ultrasound, or temperature changes [80]. Furthermore, liposomes can modify DNA by attaching specific molecular particles to their surface, making them effective anticancer agents. Proven in several experiments, their suitable biodistribution makes them highly promising tools for gene therapy and drug delivery [81].

Functionalized liposomes have molecules bound on their surface that enable them to target specific cells as well as to attach to them; that way, drug delivery to diseased tissues is improved with minimal adverse effects on healthy cells [82]. Functionalized liposomes have already been used for the delivery of antifungal drugs, antibiotics, and siRNA [83,84]. Drugs can be incorporated into liposomes through passive or active loading methods, as already mentioned. Passive loading involves trapping bioactive molecules during liposome formation, while active loading involves adding drugs to preformed liposomes [85]. 

For water-soluble drugs, encapsulation efficacy depends on the aqueous volume within the liposomes, influenced by the liposome’s morphology, phospholipid concentration, and number of lamellae. Lipophilic drugs, such as vitamin D, interact directly with the phospholipid bilayer, so encapsulation efficiency depends on the variety and concentration of phospholipids rather than morphological parameters [77,86]. Hydrophilic drugs are loaded into the liposome’s aqueous core, while hydrophobic drugs are embedded in the lipid bilayer [87]. 

Active loading involves combining preformed liposomal vesicles with concentrated drug solutions [85]. The drugs diffuse into the liposomes due to the high permeability of the phospholipid bilayer, leading to efficient encapsulation. During this process, hydrophilic drugs attach to the polar head groups of phospholipids and are entrapped within the liposomes [77]. The amount of fat-soluble drugs that can penetrate into a liposome is determined by how tightly packed the lipid bilayer is [88]. Manufacturing processes for these drugs vary significantly. Amphipathic drugs, which can diffuse through lipid bilayers, present challenges in remaining within the liposomes. Active loading offers the advantage of excluding the bioactive agent during liposome assembly, thus reducing safety risks when toxic drugs are handled. However, this method is limited to drugs that are weak amphiphilic bases or acids, which can diffuse through the bilayers only in their uncharged state [89].

The versatility of liposomes in encapsulating both hydrophilic and hydrophobic bioactive substances, either separately or together, has made it a widely favored technique for vitamin D delivery. Liposomes offer flexibility in composition and size and they demonstrate high compatibility with animal tissues as they resemble natural plasma membranes [76]. Researchers have utilized various formulations for liposomal encapsulation of vitamin D, such as 1,2-dimyristoyl-sn-glycero-3-phosphocholine (DMPC) [90], L-α-phosphatidylcholine and L-α-phosphatidyl-DL-glycerol [91], hydrogenated phosphatidylcholine [92], and soybean phosphatidylcholine [93]. The chemical stability of vitamin D within liposomes enhances its potential for food fortification [93,94]. 

A recent study presented a novel method for the effective incorporation of curcumin and vitamin D3 into powdered food formulations using liposome encapsulation and high-shear wet agglomeration with maltodextrin [95]. This approach achieved high retention rates of bioactive and improved the stability and flowability of enriched cornstarch [95]. Didar et al. showed that liposomes loaded with vitamin D3 proved effective as carriers for the fortification of dark chocolate without adversely affecting its quality attributes [96]. The chosen ethanol injection method for the preparation of vitamin D3 liposomes achieved high loading efficiency and controlled release of vitamin D3 following the Korsmeyer–Peppas model [96]. Fortified chocolate maintained color consistency, rheological properties, and sensory characteristics compared to control, with superior retention of vitamin D3 observed in liposome-fortified samples over time [96]. This study underscores the potential of liposomes to enhance the nutritional profile of chocolate products with minimal impact on their overall quality [96]. In another study, researchers successfully synthesized β-cyclodextrin/vitamin D3 (βCD/vitD3) inclusion complexes and encapsulated them within gelatin-coated nanoliposomes (NLPs) [97]. Optimization of gelatin concentration resulted in effective surface coating of the complexes, enhancing particle stability and characteristics [97]. The coated NLPs showed favorable particle size and zeta potential, along with efficient encapsulation of the βCD/VitD3 complex [97]. Furthermore, both the complex-loaded NLPs and their coated forms exhibited controlled release profiles under simulated gastrointestinal conditions, highlighting their potential as effective delivery systems for vitamin D3 supplementation [97]. An effort to optimize vitamin D3 liposomal formulation aiming to ameliorate encapsulation efficiency and to induce vitamin D transdermal absorption and stability exhibited promising results, as vitamin D3 liposomes significantly enhanced the stability and skin retention of vitamin D3 compared to its free form [98]. In a rat photoaging model, vitamin D3 liposomes improved skin morphology, promoted collagen production, and effectively repaired histological damage, indicating their potential as a superior skin care agent for prevention and treatment of photoaging [98]. Zurek et al. presented superior liposomal formulations of vitamin D3 compared to oily formulations regarding elevation in serum calcidiol concentrations [99]. The area under the curve value for the liposomal formulation, calculated from a concentration–time graph, was found to be four times greater than that of the oily formulation, indicating superior supplementation efficiency [99]. Lastly, in the study by Ghiasi et al., a novel structure of food-grade nanoliposomes stabilized by a 3D organogel network within the bilayer shell was described [100]. This structure significantly improved the stability and encapsulation efficiency of vitamin D3 [100]. The gelled bilayer shell provided enhanced stability against environmental stressors and maintained high negative zeta potential, particle size stability, and structural integrity over extended storage periods compared to control nanoliposomes without the organogel shell [100].

#### 5.1.2. Vitamin D Microencapsulation Using Nanostructured Lipid Carriers (NLCs)

NLCs were developed to enhance drug loading and stability during storage. NLCs control drug expulsion, triggered by crystallization, through their unique lipid matrix structure [101,102]. There are various classes of NLCs, including imperfect, multiple, and amorphous NLCs. Imperfect NLCs have a lipid mix with different fatty acids that create imperfections, increasing the area for drug accumulation and enhancing encapsulation efficiency [101,102]. Amorphous NLCs, made by the combination of liquid and solid lipids, form oil nano-compartments within the solid lipid phase, improving drug entrapment and stability for non-polar drugs [101,102]. Multiple NLCs include a complex mixture of different types of lipids that result in a more versatile and effective drug delivery system utilizing the benefits of multiple lipid types [101,103]. Additionally, NLCs can effectively immobilize both water-soluble and fat-soluble compounds [63]. Hybrid lipid–polymer nanoparticles, which combine the benefits of lipid and polymeric nanoparticles, are another advanced DDS [63].

Although it has the potential to lead drug delivery techniques, NLC remains one of the least-investigated methods for encapsulating vitamin D. However, its use seems promising, as it addresses challenges such as low bioavailability and stability. The combination of solid and liquid lipids in a structured matrix enables NLCs to efficiently encapsulate hydrophobic vitamin D3, leading to enhanced solubility and protection from degradation. This technology offers the advantages of controlled release and prolonged retention of vitamin D3, which can promote the development of effective nutraceutical and pharmaceutical formulations aiming to improve vitamin D supplementation strategies. Rabelo et al. showed that the use of NLCs coated with chitosan effectively encapsulated vitamin D with high stability and encapsulation efficiency (>98%) over 60 days at 25 °C, making them promising for vitamin D delivery applications [104]. The formulation NLC 70(SA):30(OA) exhibited the most favorable characteristics with minimal size variation and physical instability, pointing out its potential as a robust carrier system for vitamin D [104]. Mohammadi et al. demonstrated that NLCs formulated with Precirol demonstrated superior physical stability compared to those with Compritol, highlighting Precirol’s efficacy as a solid lipid component [93]. Optimal surfactant concentrations effectively prevented particle agglomeration during production [93]. Vitamin D3 encapsulated in NLCs showed enhanced intestinal absorption, implying that NLCs are a promising strategy for the fortification of beverages with lipophilic nutraceuticals like vitamin D [93]. In a subsequent study, the same research team examined the effectiveness of D3-NLCs formulations by using Precirol and Compritol as solid lipids, and Miglyol and octyl octanoate as liquid lipids, with tween80, tween20, and poloxamer407 as surfactants to prepare vitamin D3-loaded NLC dispersions [105]. The results showed that Precirol-based NLC demonstrated superior physical stability compared to the Compritol-based NLC [105]. An optimum concentration of 3% poloxamer407 or 2% tween20 effectively covered the nanoparticle surface, preventing agglomeration during homogenization. The study concluded that vitamin D3 incorporation into NLCs enhanced its intestinal absorption, suggesting that the use of NLCs is a promising method for fortifying beverages with lipophilic nutraceuticals, such as vitamin D [105]. 

In another trial, NLC and nanoemulsion forms of vitamin D were developed to assess their efficacy for dairy products enrichment [106]. NLC and nanoemulsion formulations of vitamin D3 exhibited features of improved bioavailability and stability, making them suitable carriers for food fortification [106]. These systems effectively protected vitamin D3 under acidic conditions, suggesting that they are capable to enrich dairy products and other fortified foods with bioactive compounds like vitamin D [106]. The effect of 1,25(OH)2D3 encapsulation in NLCs for colonic delivery via oral administration has also been explored [107]. The NLCs effectively delivered 1,25(OH)2D3 to colonic tissues, demonstrating prolonged retention and therapeutic efficacy in a murine model of colitis induced by dextran sodium sulfate [107]. This result highlighted that NLCs can be used as a promising formulation strategy for inflammatory bowel disease (IBD) treatment through targeted delivery of bioactive compounds like 1,25(OH)2D3 [107]. Junqueira et al., who evaluated the use of NLCs loaded with vitamin D3 in a transdermal emulsion, determined that these systems demonstrate excellent stability, minimal cytotoxicity, and enhanced skin permeation, highlighting their potential as a viable alternative for vitamin D supplementation [108]. In a recent study, various nanosystems for vitamin D3 loading were evaluated, underscoring again the value of NLCs as advanced delivery systems [109]. NLCs exhibited superior efficiency in loading and sustaining the release of vitamin D3, as well as great stability and the capability for extended release over one month without causing cytotoxic effects [109].

#### 5.1.3. Vitamin D Microencapsulation Using Solid Lipid Nanoparticles (SLNs)

Solid lipid nanoparticles (SLNs) are an innovative and versatile delivery system. SLNs contain solid lipid cores stabilized by surfactants and offer several advantages, including enhanced drug stability, controlled release, and improved bioavailability [110]. These nanoparticles are typically prepared using techniques such as high-pressure homogenization or solvent evaporation methods. SLNs are biocompatible and biodegradable, making them safe for various applications [111]. They effectively protect encapsulated drugs from degradation and can target delivery to specific tissues, enhancing therapeutic efficacy while minimizing side effects [110]. SLNs are particularly valuable in the delivery of poorly water-soluble drugs, proteins, and nucleic acids, offering a promising platform for advanced drug delivery solutions [112]. 

SLNs have emerged as a promising delivery system for encapsulating vitamin D due to their ability to enhance the stability, bioavailability and controlled release of lipophilic bioactive compounds. Through the incorporation of vitamin D into SLNs, the challenges associated with its poor solubility and susceptibility to degradation are effectively addressed [73]. A recent study demonstrated that the encapsulation of vitamin D with SNLs not only protected vitamin D from oxidation and thermal stress but also ensured its sustained release, making SLNs an ideal carrier for fortifying food products [113]. The beneficial role of SLNs for vitamin D3 loading has been recently assessed, highlighting their potential as an efficacious delivery system for vitamin D3 [109]. 

### 5.2. Polymers 

#### 5.2.1. Vitamin D Microencapsulation Using Polymeric Nanoparticles 

Polymeric nanoparticles are versatile carriers ranging from 1 to 1000 nm in size, available in various dosage forms like nanocapsules and nanospheres [114]. They encapsulate drugs in a liquid core of water or oil, surrounded by a solid polymeric membrane or dispersed within a polymer matrix [114]. Polymeric nanoparticles offer enhanced stability and efficient drug encapsulation, modulated by manufacturing techniques and material properties. Polymeric nanoparticles, such as nanocapsules, feature a core enclosed by a polymeric wall that can encapsulate or adsorb drugs, often functionalized for targeted delivery [115]. Biocompatible polymers are essential to nanoparticle preparation, ensuring stability, controlled biodegradation kinetics, and low toxicity upon disassembly [115]. Drug release from polymeric nanoparticles can occur via mechanisms like polymer hydration-induced expansion followed by diffusion or enzymatic degradation that trigger drug release [116]. Polymeric nanocarriers improve therapeutic efficacy considering that they both enhance drug absorption and minimize adverse effects, making them promising for diverse treatments [116].

Polymeric nanoparticles represent an advancing frontier in pharmaceutical technology, particularly for the encapsulation and delivery of hydrophobic compounds such as vitamin D. Polymeric nanoparticles offer a versatile platform due to their ability to encapsulate vitamin D within a polymeric matrix, providing protection against environmental factors and enhancing bioavailability. This approach not only ensures controlled release kinetics but also enables targeted delivery that can potentially minimize dosage frequency and adverse effects associated with conventional vitamin D formulations. The potential use of polymeric nanospheres (TyroSpheres) as a carrier for cholecalciferol topical delivery aiming to ameliorate skin delivery and stability of vitamin D has been evaluated [117]. TyroSpheres seem promising carrier systems to enhance the topical delivery and stability of vitamin D3. They exhibit high drug loading and binding efficiency of vitamin D3 within their structure, resulting in sustained release and effective penetration through the stratum corneum [117]. An ex vivo skin distribution study revealed significant delivery of active vitamin D3 into the epidermis compared to control vehicle [117]. Moreover, encapsulation in TyroSpheres did not compromise the biological activity of vitamin D3 in keratinocytes, as confirmed by in vitro cytotoxicity assays [117]. Additionally, TyroSpheres exhibited the ability to protect vitamin D3 from hydrolysis and photodegradation, highlighting their potential to improve the stability of vitamin D3 in topical formulations intended for enhanced skin delivery [117]. Jung et al. prepared polymeric nanoparticles containing ovalbumin (OVA) and active VD3 (NP[OVA+aVD3]) and assessed their role in the immunomodulatory process [118]. The results showed that both nanoparticles presented potent immunomodulatory effects; treatment with NP(OVA+aVD3) induced a tolerogenic phenotype in dendritic cells [118]. In vivo experiments further supported the immune-suppressive role of NP(OVA+aVD3), showing suppression of OVA-specific cytotoxic T lymphocytes after intravenous injection and induction of oral tolerance in mice [118]. These findings underscore the potential value of biodegradable nanoparticles as a novel strategy for antigen-specific immune modulation, implicating their future application in autoimmune disease therapy and beyond [118].

#### 5.2.2. Vitamin D Microencapsulation Using Poly L-Lactide-Co-Glycolide Nanoparticles (PLGAs)

Poly(L-lactide-co-glycolide) (PLGA) nanoparticles represent a versatile delivery system in biomedical and pharmaceutical applications [119]. Composed of biodegradable and biocompatible copolymers of lactic and glycolic acid, PLGA nanoparticles are renowned for their ability to encapsulate a variety of drugs, proteins, genes, and other bioactive molecules [119]. They are typically prepared using methods like nanoprecipitation or emulsion–solvent evaporation, with equipment such as high-pressure homogenizers, ultrasonicators, and rotary evaporators, which enable precise control over particle size, surface characteristics, and drug release kinetics [120]. Once administered, PLGA nanoparticles undergo hydrolytic degradation into non-toxic metabolites (lactic and glycolic acid), which are naturally eliminated from the human body [121]. This property not only enhances their safety profile but also facilitates controlled and sustained release of encapsulated agents, thus optimizing therapeutic efficacy while minimizing systemic side effects. PLGA nanoparticles have been extensively investigated for targeted drug delivery in cancer therapy, vaccine development, gene therapy, and regenerative medicine, highlighting their significant potential role in the improvement of healthcare solutions through tailored and effective drug delivery systems [122].

PLGA has emerged as an effective carrier system for the delivery of various therapeutic agents, including vitamin D. PLGA nanoparticles can enhance stability, prolong circulation time, and improve targeted delivery to specific tissues or cells. This makes them great candidates for addressing challenges associated with vitamin D supplementation, such as stability issues and bioavailability enhancement. In this setting, Cristelo et al. demonstrated that vitamin D3 PLGA is a very efficacious delivery system for vitamin D3 [109]. Khodaverdi et al. suggested that the co-delivery of the chemotherapy drug paclitaxel (PTX) and vitamin D3 nanoparticles using PLGA significantly enhanced the anticancer effects against MCF-7 breast cancer cells, leading to increased apoptosis, reduced cell migration, and sustained drug release [123]. These findings indicate that PLGA-based co-delivery NPs are more effective than PTX alone in the induction of primary apoptosis and the reduction of metastasis and drug toxicity [123]. Another study highlighted the efficient delivery of calcitriol into cancer cells using PLGA nanoparticles, significantly enhancing its inhibitory effects on cell growth in pancreatic and lung cancer cell lines, with reduced impact on non-tumor cells [124]. These findings suggest that PLGA nanoparticles can be used as a delivery system for calcitriol in cancer treatment, as they present enhanced efficacy and selectivity.

### 5.3. Micelles

#### Vitamin D Microencapsulation Using Micelles

Micelles are nano-sized, spherical colloidal particles with a polar outer surface and a non-polar interior [125]. They can carry bioactive agents within their hydrophobic core or bound to their surface [126]. A key advantage is their ability to quickly deliver fat-soluble medications. Micelles are formed through self-aggregation of amphiphiles in water just above their critical concentration, encapsulating fat-soluble compounds. If diluted below this concentration, they break apart and release the drug [71]. Conversely, certain blood compounds can increase the critical micelle concentration (CMC). When the CMC is increased, the existing concentration of amphiphiles may become insufficient to maintain micelle stability, effectively reducing the number of stable micelles and potentially leading to their destabilization and premature drug release [127].

Encapsulation of vitamin D in micelles represents a cutting-edge approach to induce the delivery and bioavailability of this critical fat-soluble nutrient. Micelles provide an efficient vehicle for vitamin D transport through aqueous environments throughout the body. This innovative method utilizes the unique ability of micelles to be formed spontaneously in water above a certain concentration, encapsulating vitamin D within their core and protecting it from degradation caused by oxidation, UV light, and heat [128,129]. Additionally, micelles facilitate the targeted release of vitamin D, improving its absorption and effectiveness [128,129]. This encapsulation technology can potentially address vitamin D deficiencies and enhance the efficacy of dietary supplements and pharmaceutical formulations. A randomized, double-blind clinical trial evaluated the bioavailability and safety of two vitamin D3 formulations, the standard vitamin D and the micellar vitamin D [130]. The study concluded that micellar vitamin D3 significantly improved bioavailability compared to standard vitamin D3 at a daily dose of 1000 IU, but not at 2500 IU. Both formulations were found to be safe with no significant adverse events [130]. An in vitro study compared the capability of mixed micelles as a delivery system for the improvement of vitamin D bioaccessibility to that of oil-in-water emulsions [131]. The results showed that mixed micelles significantly enhanced the bioaccessibility of vitamin D (up to 93% bioaccessibility) compared to oil-in-water emulsions, highlighting their efficiency as delivery vehicles for vitamin D in food fortification strategies [131]. Another study on mixed micelles showed that their formulation and stability were affected by the type of fatty acid, the concentration of phospholipids, and the presence of salts [132]. The incorporation of vitamin D3 in mixed micelles was consistent regardless of fatty acid type, implying the improvement of vitamin D delivery through food structure manipulation [132]. 

The study by Lowewn et al. showed that re-assembled casein micelles could effectively optimize fat-soluble vitamin loading, especially that of vitamin D, under controlled conditions [133]. These micelles demonstrated superior stability of vitamin D during dry storage compared to control powders, suggesting potential benefits for the enhancement of vitamin D stability in dry formulations [133]. Another study on re-assembled casein micelles highlighted their use as a promising natural delivery system [134]. The study showed that re-assembled casein micelles could effectively protect vitamin D3 during simulated digestion, significantly improving its retention compared to free vitamin D [134]. This protection was attributed to vitamin–casein binding and natural gelation near the casein isoelectric point, which shielded the vitamin from degradation [134]. Although absorption by Caco-2 cells was similar between digested re-assembled casein micelles and free vitamin D, the enhanced stability of re-assembled casein micelles resulted in a fourfold higher bioavailability of vitamin D3 [134]. 

### 5.4. Nanoemulsion

#### Vitamin D Microencapsulation Using Nanoemulsion Technology 

Nanoemulsion technology utilizes small particles ranging from 50 to 500 nm to create semitransparent, low-viscosity solutions [135]. It employs oil, water, and surfactants or co-surfactants to form stable formulations [135]. Compared to microemulsions, nanoemulsions require lower concentrations of surfactants, which are approved for human use [136]. These formulations can form lamellar phases, creating a thin liquid coating around the nanoemulsion droplets that enhances drug delivery. Nanoemulsions are versatile in the development of creams, liquid solutions, and foams, particularly beneficial for solubilizing lipophilic drugs and improving their bioavailability and absorption [137]. They protect drug molecules from light, enzymatic, or oxidative degradation and facilitate enhanced cellular uptake of bioactive compounds. This technology enables controlled delivery of drugs with diverse chemical properties to specific cells or tissues that positively affects drug solubility and bioavailability upon oral administration [137].

Nanoemulsion technology has revolutionized the encapsulation of vitamin D, overcoming challenges related to poor solubility and susceptibility to degradation. Kadappan et al. showed that nanoemulsion-based delivery systems significantly enhanced the bioavailability of vitamin D3 compared to coarse emulsions and vehicle controls in animal models [138]. This approach seems promising for the improvement of vitamin D status as it increases serum 25(OH)D3 levels, suggesting potential applications for human supplementation strategies [138]. Another study showed that mixed surfactant-based nanoemulsions effectively delivered vitamin D into food and beverages, thus addressing challenges of poor solubility and oxidation [139]. These nanoemulsions exhibited stable droplet sizes and retained significant amounts of vitamin D during storage at both 4 °C and 25 °C over a 2-month period [139]. The robust stability against various environmental conditions further supports their potential as a viable strategy for fortifying food products and beverages [139]. Formulated gummies with optimized texture and sensory acceptance constitute another approach that could effectively improve patient compliance when addressing vitamin D deficiency [140]. In particular, vitamin D oil-in-water nanoemulsions effectively enhanced stability and bioavailability in edible gummies, maintaining over 97% of active vitamin D over a 45-day storage period [140]. These nanoemulsions prevented degradation compared to conventional oil solutions and contributed to sustained intestinal release [140]. Nanoemulsions of vitamin D have also presented advantages in the treatment of vitamin D deficiency in animal models suffering from cerebral ischemia [141]. More specifically, vitamin D3-loaded nanoemulsions demonstrated efficient permeation across nasal mucosa and enhanced deposition in the brain implicating their potential role in the treatment of cerebral ischemia in rats [141]. Gamma scintigraphy and magnetic resonance imaging confirmed superior efficacy compared to intravenous administration of vitamin D3 solution. These findings highlight the intranasal route as effective for the delivery of vitamin D3 in settings of cerebral ischemia [141]. 

## 6. Conclusions

Recent advances in the use of organic nanocarriers for the delivery of vitamin D and its analogues have expanded the potential therapeutic applications of vitamin D. These nanocarriers, particularly lipid-based nanoparticles, have been proved effective in the enhancement of the bioavailability and therapeutic efficacy of vitamin D formulations. The interest in vitamin D and its derivatives has emerged due to their numerous beneficial effects, urging their incorporation into advanced DDSs. Recent reviews have discussed various technological strategies developed for vitamin D delivery, although many are still in early research stages. 

Lipid-based nanocarriers have garnered significant attention for their ability to stabilize vitamin D, improve its solubility, and enhance its absorption into the bloodstream. They offer advantages over conventional formulations by providing sustained release, targeting specific tissues, and increasing overall bioavailability. Self-assembled lipid nanoparticles, in particular, have demonstrated efficacy in oral delivery of vitamins, overcoming limitations such as poor solubility and permeation across epithelial barriers (Figure 4). The synthesis of low calcemic vitamin D analogues combined with targeted delivery via functionalized DDSs seems promising. 

Up to today, several groundbreaking methods have been introduced for vitamin D delivery that significantly enhance stability, bioavailability, and efficacy, with the most successful methods summarized below. More specifically, liposome encapsulation and high-shear wet agglomeration with maltodextrin [96] have demonstrated effective stabilization and controlled release of vitamin D. In addition, a novel food-grade nanoliposome structure stabilized by a 3D organogel network within the bilayer shell has shown improved delivery efficacy [100]. NLCs coated with chitosan [104] and NLCs loaded with vitamin D3 in a transdermal emulsion [108] also represent significant progress in this field. Other promising approaches include the use of polymeric nanospheres for topical cholecalciferol delivery [117] and the co-delivery of paclitaxel and vitamin D3 nanoparticles using PLGA, which has enhanced anticancer effects [123]. Micellar vitamin D3 formulations [130] and re-assembled casein micelles [133], which optimize fat-soluble vitamin loading, further contribute to the advancements in delivery methods. Moreover, mixed-surfactant-based nanoemulsions have effectively delivered vitamin D into food and beverages [139], while formulated gummies with optimized texture and sensory acceptance offer another innovative approach to vitamin D delivery [140].

In conclusion, recent advancements in organic nanocarriers for vitamin D delivery could further improve therapeutic strategies. These developments suggest a promising direction for future research and applications in optimization of vitamin D delivery. Further research in the field may enhance the stability and bioavailability of vitamin D and pave the way for personalized medicine approaches tailored to specific health conditions and patient needs. Continued research and development in this area could potentially exploit the advantages of nanotechnology and translate them into practical solutions for optimizing vitamin D therapy and beyond.

## Figures and Tables

**Figure 1 biomolecules-14-01090-f001:**
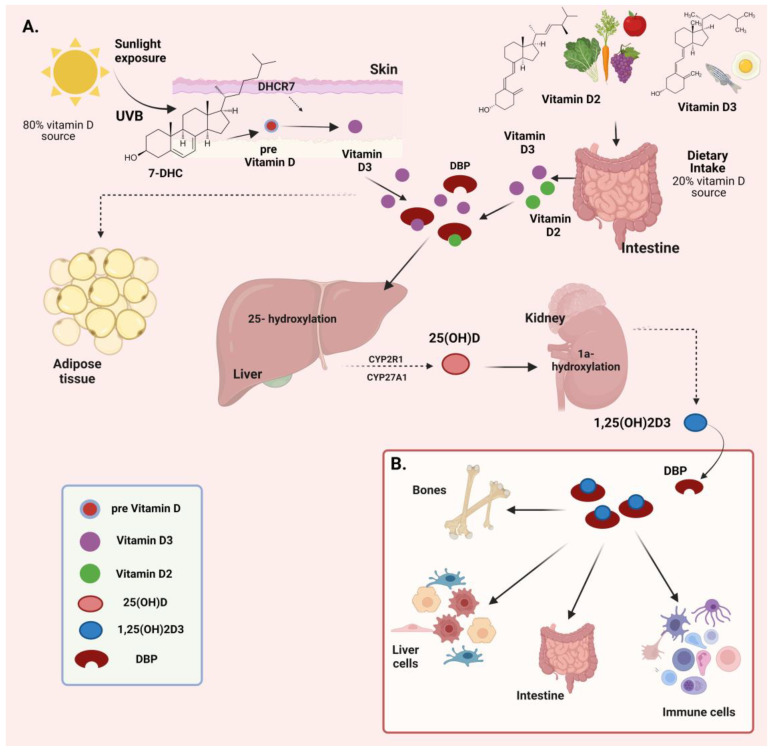
(**A**) Vitamin D synthesis and metabolism. (**B**) Calcitriol transportation to target tissues. Created with BioRender.com (accessed on 2 August 2024). Abbreviations: UVB—ultraviolet B; 7-DHC—7-dehydrocholesterol; DHCR7—7-dehydrocholesterol reductase; DBP—vitamin D-binding protein; CYP2R1—cytochrome P450, family 2, subfamily R, polypeptide 1; CYP27A1—cytochrome P450, family 27, subfamily A, polypeptide 1; 25(OH)D—25-hydroxyvitamin D; 1,25(OH)2D3—1,25-dihydroxyvitamin D3.

**Figure 2 biomolecules-14-01090-f002:**
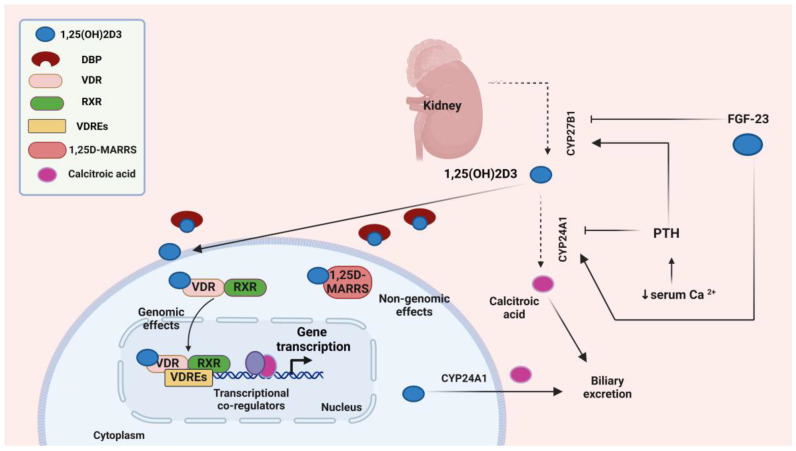
Vitamin D–Vitamin D receptor (VDR)-mediated signaling. Created with BioRender.com (accessed on 02 August 2024). Abbreviations: 1,25(OH)2D3—1,25-dihydroxyvitamin D3; DBP—vitamin D-binding protein; CYP27B1—cytochrome P450, family 27, subfamily B, polypeptide 1; CYP24A1—cytochrome P450, family 24, subfamily A, member 1; VDR—vitamin D receptor; RXR—retinoid X receptor; VDRE—vitamin D response element; 1,25D-MARRS—1,25D-membrane-associated rapid response steroid-binding protein; FGF-23—fibroblast growth factor 23; PTH—parathyroid hormone; Ca—calcium.

**Figure 3 biomolecules-14-01090-f003:**
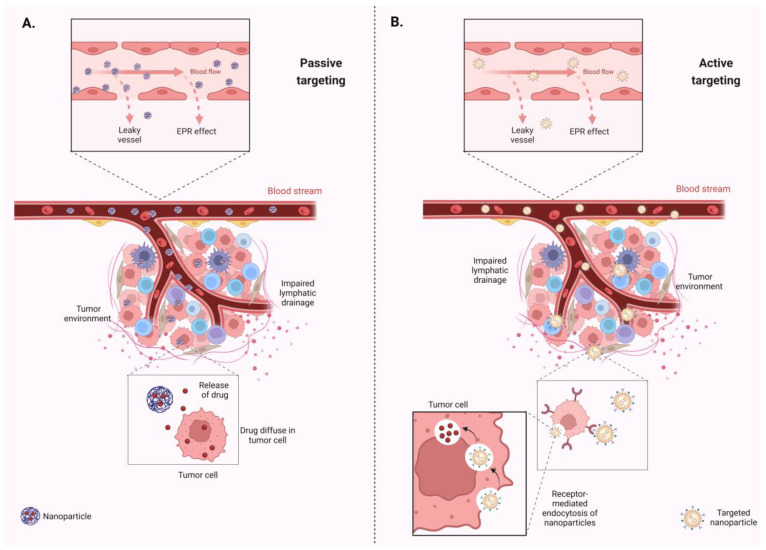
Passive and active targeting in nano-delivery systems within the tumor environment. (**A**) Passive targeting is facilitated through the enhanced permeability and retention (EPR) effect, where nanocarriers circulate through the bloodstream, permeate the leaky vasculature of tumors, and accumulate within the tumor tissue. (**B**) Active targeting involves modification of nanocarriers with specific targeting ligands that recognize and bind to receptors highly expressed on tumor cells. This targeted approach facilitates local drug delivery and internalization through receptor-mediated endocytosis, thereby enhancing the efficacy of anti-tumor therapies. Created with BioRender.com (accessed on 2 August 2024). Abbreviations: EPR—enhanced permeability and retention.

**Figure 4 biomolecules-14-01090-f004:**
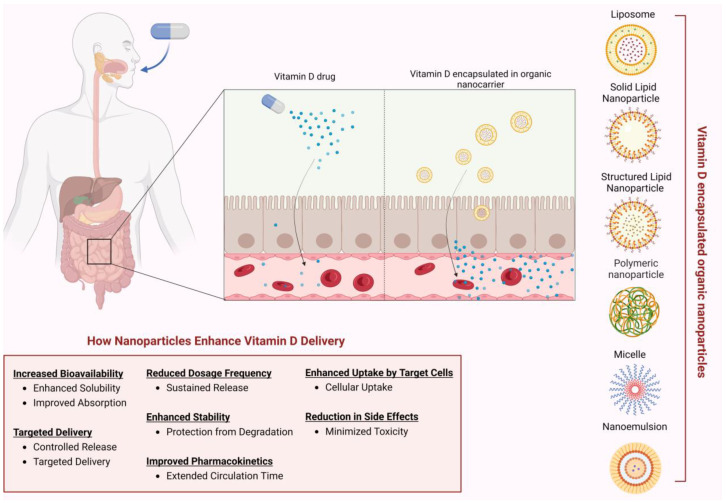
Types of nanoparticles for vitamin D delivery, enhanced intestinal absorption of vitamin D encapsulated nanoparticles compared to free drugs, and mechanisms of improved delivery and effectiveness. Created with BioRender.com (accessed on 2 August 2024).

**Table 1 biomolecules-14-01090-t001:** Advantages and disadvantages of each delivery platform for vitamin D.

Delivery Platform	Advantages	Disadvantages
**Lipid-based Nanoparticles**	- Improved stability - Enhanced solubility - Targeted delivery - Sustained release - Increased bioavailability	- Potential biocompatibility issues - Manufacturing complexity - Higher production cost
**Polymeric Nanoparticles**	- Enhanced stability - Prolonged circulation time - Targeted delivery to specific tissues - Improved bioavailability	- Potential for immune response - Biocompatibility challenges - Complex manufacturing processes
**Micelles**	- Efficient transport through aqueous environments - Protection from degradation - Improved absorption - Enhanced bioaccessibility - Stability during digestion	- Variable formulation stability depending on fatty acids and salts - Potential absorption variability
**Re-assembled Casein Micelles**	- Superior stability during dry storage - Enhanced retention during digestion - Higher bioavailability compared to free Vitamin D	- Potential complexity in formulation - Requirement for controlled conditions for effective delivery
**Nanoemulsions**	- Enhanced bioavailability - Stable droplet sizes - Significant retention during storage - Effective against various environmental conditions - Potential for treating cerebral ischemia	- Require specific surfactants - Potential for stability issues over long periods - Formulation complexity

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
