# Peer review of "Recent Advances in the Use of Vitamin D Organic Nanocarriers for Drug Delivery"

_biomolecules, 2024, doi:10.3390/biom14091090_

Round 1

Reviewer 1 Report

Comments and Suggestions for Authors

The beginning of the paper (sections 1-4) spends a great deal of time explaining the body’s use and metabolism of vitamin D, causing the first seven pages to lack novelty and relevance to advances in nanocarriers. While sparse locations within the text do hint at the significance of vitamin D due to its potential ability to improve numerous body functions and disorders, the beginning can be improved by explaining more about vitamin D’s newly discovered roles and need for nanoformulation. As it stands this text does not bring to light any new topics or scientific research, as these are all elements that can be found in a biology textbook.

Furthermore, the section on drug-delivery systems can be improved by providing statistical evidence that shows a numerical comparison between how much of a drug reaches its target site with and without nanocarriers.

The figures require revisions as they are disorganized with too much information in a small amount of space. They do not correlate well with the text discussion.

The paper can be improved by mentioning some of the methods of successful trials referenced.

The strengths of the paper begin on page 8, where the content is organized by the type of particle, chart of applications for that specific class of particle, and respective results regarding transport, release, and encapsulation efficiency.

However, the therapeutic efficacy of vitamin D once encapsulated or released is not addressed. This observation is highlighted in the conclusions (p. 15 line 627) which present a ‘revolution in therapeutic strategies’ or of “anti-cancer therapies” (line 625). Indeed, the results of the studies cited above do not support these conclusions.

The abstract needs to be re-written. The ending is weak and needs to provide a rationale as to the importance of this review to the readership of biomolecules. It is not clear what the various forms of vitamin D to which the authors are referring (line 18). See additional comments below on use of the words emerging/potential/prospects. It is also not clear what the types of organic nanocarriers the authors will discuss in the article from the abstract.

Page 1: Remove the period from the end of the title

Abstract

Page 1, line 11-12: Nanotechnology is no longer considered an emerging field. It is not a new concept but rather now considered a general-purpose technology that has revolutionized industry. (see DOI: 10.3390/molecules28020661 and others) Re-motivation should be considered in this sentence and removal of the words “potential” and “prospects”. It is not clear how nanotechnology provides “secure” drug delivery. This is likely the wrong word choice.

Section 1 Introduction

Page 1, line 27: “Nowadays” is too colloquial.

Page 1, lines 31 & 35, Page 3 line 102, Page 4 line 147: spacing is off, extra spaces present

Page 1, line 31: “in parallel” is the wrong word choice and does not match the connection with the previous sentence, the phrase “for example” would be better suited.

Page 1, line 35-36: add example genes that are regulated by VDR binding with calcitriol. “involved in cell growth (e.g., BCL2,

It should also be noted that calcitriol can act independently of VDR, by rapidly activating signaling molecules such as PLC, PI3K and MAP kinases

Page 1, line 37: modify twice usage of pathogenic/pathogenesis in the same sentence.

Page 1, line 39: provide the types of cancers studied with references for each

Page 1, line 39: modify usage of “in parallel” if not changed on line 31

Page 1, line 42-45: run-on sentence

Page 2, line 48: introduce 25(OH)D as ergocalciferol prior to acronym usage

Page 2, line 51: it is not necessary to continue to use both insufficiency and deficiency; one is sufficient

Page 2, lines 53-61: the previous paragraph speaks on vitamin D deficiency and this paragraph (lines 53-61) switches the discussion to vitamin D toxicity; to enhance logic and organization, the authors should explain the discussion on toxicity immediately after deficiency. The transition is very disjointed and abrupt. The authors then discuss low bioavailability and back to deficiency, the flow is not logical.

Page 2, line 72: referring to passive and active targeting mechanisms is too vague here and needs to be elaborated upon. Include the types of active targeting used in delivery of Vitamin D.

Page 2, line 82-87: It is still not clear the types of organic nanocarrier systems (polymeric, lipids, etc.?) that will be discussed. Also, what “other materials” are combined with Vitamin D. The ending is vague and weak.

Section 2

Page 3, Figure 1: figure 1 is crowded with too much small and detailed information that not legible. Separating the steps in figure 1 after the kidney converts 25(OH)D into 1,25(OH)2D3 which needs to be defined in the abbreviations as 1,25-dihydroxyvitamin D3 into a separate figure regarding intracellular pathways would be beneficial. Why show adipose tissue, intestine and immune cells if not introduced when the figure is discussed (lines 92, 108, 111, 116, 119, 122, 124, 134, 155)? Fig 1 is referenced eight different times in the text. Each reference should be labeled and divided into sections A-F (ie. Fig 1A). Often the text placement refering to Figure 1 is awkward as the figure is not presenting any information about food sources.

Page 3, line 94: should be “and Vitamin D receptor (VDR)”

Page 3, line 105: according to the text, the major forms of Vitamin D have different chemical structures, but these chemical differences and difference in binding affinity are not discussed. The authors only state that they are different in terms of origin/synthesis. The same is also not apparent in Figure 1. It would be better to present the chemical structures of vitamin D2 and D3 along with a discussion about the differences in structure and sources.

Page 3, Line 106: no comma is needed after ergocalciferol.

Page 4, line 116: change vitamin D-DBP to just DBP.

Page 4, lines 130 to 160:  Since the various studies cited below are not used to support/address the interaction between the various receptors, enzymes, hormones, etc., the molecular biology background is superfluous. This text needs to be better incorporated into the topic of nanocarriers and drug delivery. Context is needed for the introduction of cAMP and NR4A2 transcription.

Page 4, line 134: remove “the” before “metabolism”

Page 4, line 137: “despite” is not the correct word choice and is overused as a conjecture (see lines 130 and 371)

Page 4, line 142: although FGF-23 is mentioned in the abbreviations on page 3, it should also be defined when first introduced in the text.

Page 4, line 146 & Page 5, line 181: two sentences does not make a paragraph

Section 3

Page 5, line 165-166: stating that “VDR functions depend on its molecular structure” should be followed by specific examples of how its structure affects its function. The following sentences do not sufficiently explain specific aspects of VDR’s molecular structure that relate to its function

Section 4

Page 5, line 187-200 This paragraph is not related to drug delivery systems

Section 5

Page 6, Figure 2: Split into a A and B. On the active targeting side of figure two, the enlarged image beside the phrase “receptor mediated endocytosis of nanoparticles” needs a different origin, meaning the dotted lines do not seem to be coming from the correct location. Fig. 2 is referenced three different times in the text.

Page 9, line 324: the “ethanol injection method” needs to be elaborated upon, where is the ethanol being injected?

Page 10, line 347: it might be best to state the axes of the graph whose integral was calculated

Page 13, line 492: When describing the preparation of PLGA nanoparticles, include the machines that “enable precise control over particle size, surface characteristics, and drug release kinetics”.

Page 13, line 510: Since the paper mainly speaks on vitamin D and nanocarriers, readers may not be familiar with chemotherapy drugs. Therefore, I recommend informing readers that paclitaxel is a chemotherapy drug

Page 13, line 528: Since the previous sentence says that a decrease in concentration causes micelles to break apart and release the drug, please explain how an increase in concentration mentioned in this sentence will cause the same outcome

Page 14, line 560: please specifically define what the micelles are protecting vitamin D from

Comments on the Quality of English Language

English editing is needed related to word choice, run-ons, and organization of topics. The paragraphs do not have clear topic sentences, development and support, and conclusions. The discussion sections are not logical, organized, or properly motivated. (See additional comments)

Author Response

Responses to the Reviewers’ comments

Reviewer 1

Comment 1: The beginning of the paper (sections 1-4) spends a great deal of time explaining the body’s use and metabolism of vitamin D, causing the first seven pages to lack novelty and relevance to advances in nanocarriers. While sparse locations within the text do hint at the significance of vitamin D due to its potential ability to improve numerous body functions and disorders, the beginning can be improved by explaining more about vitamin D’s newly discovered roles and need for nanoformulation. As it stands this text does not bring to light any new topics or scientific research, as these are all elements that can be found in a biology textbook.

Response to comment 1: We understand the concern of reviewer 1 about the initial sections of the paper focusing extensively on the body's use and metabolism of vitamin D, which may seem to lack novelty and direct relevance to advances in nanocarriers. Our intention with the detailed explanation of vitamin D metabolism was to establish a comprehensive background for readers who may not be familiar with this aspect, thereby providing a solid foundation for understanding the subsequent discussions on nanocarriers. In addition, in the first seven pages of the manuscript, two figures are included. The first figure illustrates the synthesis, metabolism, and signaling pathways involving vitamin D, which is essential background information for readers. The second figure describes passive and active targeting mechanisms in nano-delivery systems within the tumor microenvironment. This information is crucial for the subsequent sections of the review, particularly when discussing how drugs are incorporated into nanocarriers.

To address Reviewer’s concerns, we have shortened some sections which may contain redundant information, as suggested. Moreover, we have enriched the manuscript with the recent research and discoveries about vitamin D’s roles, particularly those that underscore its therapeutic potential and need for advanced delivery methods (lines 49-56).  In the manuscript there is already information on the limitations of conventional vitamin D delivery systems, highlighting the necessity for nanoformulation (lines 57-70).

Comment 2: Furthermore, the section on drug-delivery systems can be improved by providing statistical evidence that shows a numerical comparison between how much of a drug reaches its target site with and without nanocarriers.

Response to comment 2: We have added real-world data in the Drug Delivery Systems Section, as suggested by the reviewer (lines 265-280).

Comment 3: The figures require revisions as they are disorganized with too much information in a small amount of space. They do not correlate well with the text discussion.

Response to comment 3: All the changes suggested by the reviewer in his comments regarding figures, figure legends, and related aspects have been addressed in detail.

Comment 4: The paper can be improved by mentioning some of the methods of successful trials referenced.

Response to comment 4: We have added a paragraph in the conclusion section describing several successful methods referenced in the trials (685-701).

Comment 5: The strengths of the paper begin on page 8, where the content is organized by the type of particle, chart of applications for that specific class of particle, and respective results regarding transport, release, and encapsulation efficiency.

Response to comment 5: As mentioned above, our introductory sections aimed to establish a comprehensive background for readers who may not be familiar with this subject, thereby providing a solid foundation for understanding the subsequent discussions on nanocarriers.

Comment 6: However, the therapeutic efficacy of vitamin D once encapsulated or released is not addressed. This observation is highlighted in the conclusions (p. 15 line 627) which present a ‘revolution in therapeutic strategies’ or of “anti-cancer therapies” (line 625). Indeed, the results of the studies cited above do not support these conclusions.

Response to comment 6: We have revised the conclusion section to better reflect the current status of vitamin D drug delivery systems in the scientific field (lines 702-710).

Comment 7: The abstract needs to be re-written. The ending is weak and needs to provide a rationale as to the importance of this review to the readership of biomolecules. It is not clear what the various forms of vitamin D to which the authors are referring (line 18). See additional comments below on use of the words emerging/potential/prospects. It is also not clear what the types of organic nanocarriers the authors will discuss in the article from the abstract.

Response to comment 7: The abstract has been re-written according to reviewer suggestions (lines 12-25).

Reviewer 1 _ Further Comments

Page 1: Remove the period from the end of the title

Response to comment: The period at the end of the title has been removed.

Abstract: Page 1, line 11-12: Nanotechnology is no longer considered an emerging field. It is not a new concept but rather now considered a general-purpose technology that has revolutionized industry. (see DOI: 10.3390/molecules28020661 and others) Re-motivation should be considered in this sentence and removal of the words “potential” and “prospects”. It is not clear how nanotechnology provides “secure” drug delivery. This is likely the wrong word choice.

Response to comment: The abstract has been re-written according to reviewer suggestions (lines 12-25).

 Section 1 Introduction

Page 1, line 27: “Nowadays” is too colloquial.

Response to comment: Nowadays has been removed.

Page 1, lines 31 & 35, Page 3 line 102, Page 4 line 147: spacing is off, extra spaces present

Response to comment: Extra spaced have been removed.

Page 1, line 31: “in parallel” is the wrong word choice and does not match the connection with the previous sentence, the phrase “for example” would be better suited.

Response to comment: The term “in parallel” has been replaced with “for example,” as suggested by the reviewer (line 35).

Page 1, line 35-36: add example genes that are regulated by VDR binding with calcitriol. “involved in cell growth (e.g., BCL2,

Response to comment: We have provided examples of genes involved in cell differentiation, immune function, and inflammation, as suggested by the reviewer (lines 41-43).

It should also be noted that calcitriol can act independently of VDR, by rapidly activating signaling molecules such as PLC, PI3K and MAP kinases

Response to comment: We have incorporated this information in Section 3, where the non-genomic pathways of vitamin D are described (lines 201-204).

Page 1, line 37: modify twice usage of pathogenic/pathogenesis in the same sentence.

Response to comment: We have revised as suggested (line 44).

Page 1, line 39: provide the types of cancers studied with references for each

Response to comment: Vitamin D and its polymorphisms have been linked with a wide variety of cancer types, including but not limited to colorectal, breast, prostate, ovarian, lung, pancreatic, liver, and hematologic cancers. However, detailing the specific types of cancers studied along with their references may extend beyond the focus of the current review.  

Page 1, line 39: modify usage of “in parallel” if not changed on line 31

Response to comment: The usage of “in parallel” has been removed.

Page 1, line 42-45: run-on sentence

Response to comment: The sentence has been revised as suggested (lines 49-53).

Page 2, line 48: introduce 25(OH)D as ergocalciferol prior to acronym usage

Response to comment: We thank the reviewer for the comment.  However, we are not entirely certain about the specific concern raised. We would like to clarify that ergocalciferol and 25-hydroxyvitamin D (25(OH)D) are not the same. Ergocalciferol is a form of vitamin D2, which is a precursor that, upon ingestion, undergoes metabolism in the liver to produce 25-hydroxyvitamin D (25(OH)D).  

Page 2, line 51: it is not necessary to continue to use both insufficiency and deficiency; one is sufficient

Response to comment: In response to Major Comment 1, the sentence in question has been removed.

Page 2, lines 53-61: the previous paragraph speaks on vitamin D deficiency and this paragraph (lines 53-61) switches the discussion to vitamin D toxicity; to enhance logic and organization, the authors should explain the discussion on toxicity immediately after deficiency. The transition is very disjointed and abrupt. The authors then discuss low bioavailability and back to deficiency, the flow is not logical.

Response to comment: We acknowledge that the transition from discussing vitamin D deficiency to its potential toxicity and related challenges might seem abrupt. To enhance the logic and organization of the discussion, we have revised the paragraph to create a smoother transition between these topics (lines 57-70).

Page 2, line 72: referring to passive and active targeting mechanisms is too vague here and needs to be elaborated upon. Include the types of active targeting used in delivery of Vitamin D.

Response to comment: The types of active targeting for vitamin D delivery, including nanotechnology and beyond, have been added, as suggested by the reviewer (lines 260-264).

Page 2, line 82-87: It is still not clear the types of organic nanocarrier systems (polymeric, lipids, etc.?) that will be discussed. Also, what “other materials” are combined with Vitamin D. The ending is vague and weak.

Response to comment: We have rewritten the aim of the study focusing into the reviewer’s suggestions (lines 99-107).

Section 2 Page 3, Figure 1: figure 1 is crowded with too much small and detailed information that not legible. Separating the steps in figure 1 after the kidney converts 25(OH)D into 1,25(OH)2D3 which needs to be defined in the abbreviations as 1,25-dihydroxyvitamin D3 into a separate figure regarding intracellular pathways would be beneficial. Why show adipose tissue, intestine and immune cells if not introduced when the figure is discussed (lines 92, 108, 111, 116, 119, 122, 124, 134, 155)? Fig 1 is referenced eight different times in the text. Each reference should be labeled and divided into sections A-F (ie. Fig 1A). Often the text placement refering to Figure 1 is awkward as the figure is not presenting any information about food sources.

Response to comment: We have divided Figure 1 into two figures. Moreover, the revised Figure 1 has been further divided into two parts, A and B, according to the reviewer’s suggestions. All references to Figure 1 in the revised manuscript have been updated to reflect the new Figure 1 and Figure 2. Additionally, the necessary changes have been made where appropriate, as per the reviewer’s suggestions. Regarding the adipose tissue, intestine and immune cells, the relevant references have been included in the revised text (lines 130-132 & 141-143).

Page 3, line 94: should be “and Vitamin D receptor (VDR)”

Response to comment: We have revised as suggested (line 155).

Page 3, line 105: according to the text, the major forms of Vitamin D have different chemical structures, but these chemical differences and difference in binding affinity are not discussed. The authors only state that they are different in terms of origin/synthesis. The same is also not apparent in Figure 1. It would be better to present the chemical structures of vitamin D2 and D3 along with a discussion about the differences in structure and sources.

Response to comment: A brief discussion on the chemical structures of vitamin D2 and D3, along with their relative sources, has been added to the revised manuscript (lines 118-128). Additionally, this information has been incorporated into Figure 1.

Page 3, Line 106: no comma is needed after ergocalciferol.

Response to comment: The comma has been removed.

Page 4, line 116: change vitamin D-DBP to just DBP.

Response to comment: We have revised as suggested (line 133).

Page 4, lines 130 to 160:  Since the various studies cited below are not used to support/address the interaction between the various receptors, enzymes, hormones, etc., the molecular biology background is superfluous. This text needs to be better incorporated into the topic of nanocarriers and drug delivery. Context is needed for the introduction of cAMP and NR4A2 transcription.

Response to comment: We have revised as suggested.

Page 4, line 134: remove “the” before “metabolism”

Response to comment: The “the” has been removed.

Page 4, line 137: “despite” is not the correct word choice and is overused as a conjecture (see lines 130 and 371)

Response to comment: The word "despite" has been substituted as suggested by the reviewer (line 149 & 410).

Page 4, line 142: although FGF-23 is mentioned in the abbreviations on page 3, it should also be defined when first introduced in the text.

Response to comment: The definition of FGF is on line 152.

Page 4, line 146 & Page 5, line 181: two sentences does not make a paragraph

Response to comment: The first paragraph has been removed according to the reviewer's previous comment. The second paragraph has been merged with the preceding one, as suggested (lines 177-181).

Section 3

Page 5, line 165-166: stating that “VDR functions depend on its molecular structure” should be followed by specific examples of how its structure affects its function. The following sentences do not sufficiently explain specific aspects of VDR’s molecular structure that relate to its function

Response to comment: We have included specific examples as suggested by the reviewer, along with the appropriate references (lines 183-187).

Section 4

Page 5, line 187-200 This paragraph is not related to drug delivery systems

Response to comment: The paragraph has been removed.

Section 5

Page 6, Figure 2: Split into a A and B. On the active targeting side of figure two, the enlarged image beside the phrase “receptor mediated endocytosis of nanoparticles” needs a different origin, meaning the dotted lines do not seem to be coming from the correct location. Fig. 2 is referenced three different times in the text.

Response to comment: We thank the reviewer for this comment. We have adjusted the location of the enlarged image origin as suggested. Additionally, we have divided Figure 2 (now Figure 3 in the revised manuscript) into two parts, A and B, as recommended.

Page 9, line 324: the “ethanol injection method” needs to be elaborated upon, where is the ethanol being injected?

Response to comment: According to Didar et al., liposomes were prepared by dissolving 30 mg of phospholipids, 10 mg of cholesterol, and 10 mg of Vitamin D3 in 5 mL of ethanol, which was then rapidly injected into 200 mL of distilled water under vortex mixing for 30 seconds. The ethanol was subsequently removed by rotary evaporation, and the liposomes were concentrated to 100 mL at 35°C, followed by sonication for 5 minutes in an ice bath for particle size, zeta potential, and loading efficiency measurements. In fact, the ethanol injection method was described for the preparation of liposomes, although no actual injection was performed. However, these details are technical, and we believe it is unnecessary to include them in the text. In the revised manuscript, we have added that the injection method was used for the preparation of Vitamin D3 liposomes (lines 362-363).

Page 10, line 347: it might be best to state the axes of the graph whose integral was calculated

Response to comment: We have added this information, as suggested by the reviewer (lines 387).

Page 13, line 492: When describing the preparation of PLGA nanoparticles, include the machines that “enable precise control over particle size, surface characteristics, and drug release kinetics”.

Response to comment: We have added these data, as suggested by the reviewer (lines 539-540).

Page 13, line 510: Since the paper mainly speaks on vitamin D and nanocarriers, readers may not be familiar with chemotherapy drugs. Therefore, I recommend informing readers that paclitaxel is a chemotherapy drug

Response to comment: We have revised as suggested (lines 556-557).

Page 13, line 528: Since the previous sentence says that a decrease in concentration causes micelles to break apart and release the drug, please explain how an increase in concentration mentioned in this sentence will cause the same outcome

Response to comment: This contradiction between the sentences has been explained (lines 576-580).

Page 14, line 560: please specifically define what the micelles are protecting vitamin D from

Response to comment: We have defined the factors that micelles protect vitamin D from, as suggested by the reviewer (line 586-587).

English editing is needed related to word choice, run-ons, and organization of topics. The paragraphs do not have clear topic sentences, development and support, and conclusions. The discussion sections are not logical, organized, or properly motivated. (See additional comments).

Response to comment: Based on the reviewer’s comments, we have made significant improvements to the manuscript. We have ensured that paragraphs now have clear topic sentences, proper development, support, and conclusions. Additionally, we have performed extensive English language editing to address issues related to word choice, run-on sentences, and overall organization, using the track changes tool. We appreciate the constructive comments and believe they have greatly enhanced the quality of our manuscript.

Reviewer 2 Report

Comments and Suggestions for Authors

In my opinion this is a good work, informative for the scientists involved in the field. I have only a few suggestions:

1) Include at least one more figure

2) Do not break the paper in so many sections. Intro to the nanoplatform and its use for delivering VitD should be combined

3) MOST IMPORTANT!!!! Justify in the intro why inorganic platforms and natural carriers (i.e., extracellular vesicles) were not included.

4) make a table of the advantage and disadvantage of each delivery platform in delivering VitD

Author Response

Reviewer 2

In my opinion this is a good work, informative for the scientists involved in the field. I have only a few suggestions:

Comment 1: Include at least one more figure

Response to comment 1:  We would like to express our gratitude to the reviewer for their comments. In the revised manuscript, we have divided Figure 1 into two separate figures to enhance clarity and comprehension. Consequently, the revised manuscript now includes a total of four figures. We hope this adjustment meets the reviewer’s approval.

Αρχή φÏŒρμας

Comment 2: Do not break the paper in so many sections. Intro to the nanoplatform and its use for delivering VitD should be combined.

Response to comment 2: We have combined the introduction to the nanoplatform with its application for delivering vitamin D, as suggested by the reviewer.

 Comment 3: MOST IMPORTANT!!!! Justify in the intro why inorganic platforms and natural carriers (i.e., extracellular vesicles) were not included.

Response to comment 3:  We have added a justification in the introduction explaining why inorganic platforms and natural carriers, such as extracellular vesicles, were not included in this review (lines 94-98). The focus was intentionally placed on organic nanocarriers to provide a detailed examination of their specific design, engineering, and applications in combination with vitamin D. We acknowledge that inorganic platforms and natural carriers are also significant in the field, but the scope of this review is centered on the advancements and synergistic effects achieved with organic nanocarriers.  

Comment 4: Μake a table of the advantage and disadvantage of each delivery platform in delivering VitD.

Response to comment 4: As suggested by the reviewer, we have added Table 1, which represents the advantages and disadvantages of each delivery platform for vitamin D.

Comment 5: English language fine. No issues detected

Response to comment 5: We appreciate the constructive comments and believe they have greatly enhanced the quality of our manuscript.

Round 2

Reviewer 1 Report

Comments and Suggestions for Authors

The authors have successfully addressed numerous points requested in the first round of revision. For example, the introduction has been shortened and is less repetitive. Additionally, figure 1 has been broken into two sections and is now easier to read. Further, the authors elaborated more where requested. However, the manuscript is still not ready for publication as there are numerous grammatical and language errors that must be revised. While the topic of the manuscript is novel and interesting to the readership of this journal, its quality is not ready to proceed for publication.

Scientific corrections that still need to be performed:

-       There are two figures labeled figure 3

-       Table one groups numerous facts together without citations

Comments on the Quality of English Language

Language corrections that still need to be performed:

-       Grammar in abstract needs revising

-       Examples: Page 1 line 14 “that way” is not the correct word choice

-       Page 1 line 17 “is increasing being explored” has too many verbs

-       Page 1 line 21-32 is a run-on sentence

-       Many of the grammar revisions in the rest of manuscript (highlighted in red) are increasing the number of grammatical errors not addressing them (e.g. page 2 line 57, line 84-86 is now a run-on, among others). 

Author Response

Reviewer 2

The authors have successfully addressed numerous points requested in the first round of revision. For example, the introduction has been shortened and is less repetitive. Additionally, figure 1 has been broken into two sections and is now easier to read. Further, the authors elaborated more where requested. However, the manuscript is still not ready for publication as there are numerous grammatical and language errors that must be revised. While the topic of the manuscript is novel and interesting to the readership of this journal, its quality is not ready to proceed for publication. Scientific corrections that still need to be performed:

Comment 1: There are two figures labeled figure 3

Response to comment 1: We apologize for the oversight. Figure 4 has now been corrected with the appropriate labeling. Thank you for bringing this to our attention.

Comment 2: Table one groups numerous facts together without citations

Response to comment 2: The table was added based on a previous reviewer's request to provide a concise overview of the advantages and disadvantages of each delivery platform for Vitamin D. The purpose of this table is to present this information in a summarized form for easy reference. We intentionally chose not to include citations in the table to maintain its clarity and comprehensiveness, allowing readers to quickly compare the delivery platforms without distraction. However, we have provided detailed references and citations in the main text to support the information presented in the table.

Comment 3: Language corrections that still need to be performed:

-       Grammar in abstract needs revising

-       Examples: Page 1 line 14 “that way” is not the correct word choice

-       Page 1 line 17 “is increasing being explored” has too many verbs

-       Page 1 line 21-32 is a run-on sentence

-       Many of the grammar revisions in the rest of manuscript (highlighted in red) are increasing the number of grammatical errors not addressing them (e.g. page 2 line 57, line 84-86 is now a run-on, among others). 

 Response to comment 3: We have made the appropriate changes as suggested by the reviewer.